# FABP5 Is a Possible Factor for the Maintenance of Functions of Human Non-Pigmented Ciliary Epithelium Cells

**DOI:** 10.3390/ijms25179285

**Published:** 2024-08-27

**Authors:** Megumi Higashide, Megumi Watanabe, Tatsuya Sato, Araya Umetsu, Nami Nishikiori, Toshifumi Ogawa, Masato Furuhashi, Hiroshi Ohguro

**Affiliations:** 1Departments of Ophthalmology, Sapporo Medical University School of Medicine, Sapporo 060-8556, Japan; 2Department of Cardiovascular, Renal and Metabolic Medicine, Sapporo Medical University School of Medicine, Sapporo 060-8556, Japan; 3Department of Cellular Physiology and Signal Transduction, Sapporo Medical University School of Medicine, Sapporo 060-8556, Japan

**Keywords:** human non-pigmented ciliary epithelium cells (HNPCE), FABP5, mitochondrial respiration, glycolysis, seahorse bioanalyzer, RNA sequencing, ingenuity pathway analysis (IPA)

## Abstract

To elucidate the possible biological roles of fatty acid-binding protein 5 (FABP5) in the intraocular environment, the cells from which FABP5 originates were determined by using four different intraocular tissue-derived cell types including human non-pigmented ciliary epithelium (HNPCE) cells, retinoblastoma (RB) cells, adult retinal pigment epithelial19 (ARPE19) cells and human ocular choroidal fibroblast (HOCF) cell lines, and the effects of FABP ligand 6, a specific inhibitor for FABP5 and FABP7 were analyzed by RNA sequencing and seahorse cellular metabolic measurements. Among these four different cell types, qPCR analysis showed that FABP5 was most prominently expressed in HNPCE cells, in which no mRNA expression of FABP7 was detected. In RNA sequencing analysis, 166 markedly up-regulated and 198 markedly down-regulated differentially expressed genes (DEGs) were detected between non-treated cells and cells treated with FABP ligand 6. IPA analysis of these DEGs suggested that FABP5 may be involved in essential roles required for cell development, cell survival and cell homeostasis. In support of this possibility, both mitochondrial and glycolytic functions of HNPCE cells, in which mRNA expression of FABP5, but not that of FABP7, was detected, were shown by using a Seahorse XFe96 Bioanalyzer to be dramatically suppressed by FABP ligand 6-induced inhibition of the activity of FABP5. Furthermore, in IPA upstream analysis, various unfolded protein response (UPR)-related factors were identified as upstream and causal network master regulators. Analysis by qPCR analysis showed significant upregulation of the mRNA expression of most of UPR-related factors and aquaporin1 (AQP1). The findings in this study suggest that HNPCE is one of intraocular cells producing FABP5 and may be involved in the maintenance of UPR and AQP1-related functions of HNPCE.

## 1. Introduction

Fatty acid-binding proteins (FABPs) belong to a family of proteins that function in various fatty acid (FA)-related biological signaling pathways by transporting FAs to suitable targets of intracellular compartments [1,2]. Among the proteins in this family, FABP4 and FABP5 are expressed in not only cells in fatty tissue but also in cells in non-fatty tissues including macrophages and neurons, thereby contributing to various FA-related physiological roles in these tissues [1,2]. In addition, FABP4 and FABP5 have been shown to be involved in the pathogenesis of various diseases including diabetes mellitus (DM), hypertension (HT), hyperlipidemia (HL) [3], renal disease [4], heart diseases [5] and malignant tumors [6]. Since FABP4 and FABP5 are secreted into various bodily fluids, the levels of FABP4 and FABP5 have been identified as key pathogenic indices for these diseases [1,2].

Recently, our group found that levels of FABP4, FABP5 and FAs in vitreous fluid surgically obtained from patients with various retinal vascular diseases (RVDs), which are ocular complications associated with DM, HT and HL [7,8,9,10], were significantly increased compared with the levels in vitreous fluid obtained from patients with epiretinal membrane (non-RVD) [11]. In addition, although the vitreous levels of FABP4, FABP5 and FAs were significantly correlated with levels of vitreous vascular endothelial growth factor (VEGFA), which is known as the most critical pathogenic factor for RVDs [12], correlation analyses indicated that levels of intraocular FABP4 (ioFABP4), ioFABP5 or ioFAs were independently regulated by ioVEGFA and had not merely originated from peripheral blood [11]. Based on collective observations, we speculated that FABP4 and FABP5 are key factors in addition to VEGFA that are involved in the pathogenesis of RVDs and that targeting FABP4 and FABP5 may therefore lead to a future therapeutic strategy for RVDs. However, since ioFABP4 and ioFABP5 were also detected in patients with non-RVDs, though their levels in those patients were lower than the levels in patients with RVDs [11], it was rationally speculated that ioFABP4 and ioFABP5 may also be involved in intraocular physiology. To address this issue, we first aimed to elucidate the intraocular origin and possible physiological functions of ioFABP4 by using four different intraocular tissue-derived cell types including human non-pigmented ciliary epithelium (HNPCE) cells, retinoblastoma (RB) cells, ARPE19 cells and human ocular choroidal fibroblast (HOCF) cells [13]. Analysis by qPCR indicated that HOCF cells were the primary cells for the origin of ioFABP4. A Seahorse cellular metabolic analysis showed that BMS309403, a specific inhibitor for FABP4, induced significant reductions of mitochondrial and glycolytic functions in a dose-dependent manner. Collectively, we concluded that ioFABP4 originates from the ocular choroid and has a critical role in cellular homeostasis of HOCF cells [13].

In the present study, we performed a similar study to elucidate the unidentified origin of ioFABP5 and its possible roles in intraocular physiology. Using four different intraocularly originating cells, HNPCE cells, RB cells, ARPE19 cells and HOCF cells, and FABP ligand 6, a specific FABP5 and FABP7 inhibitor, we carried out qPCR analysis for FABP5, FABP7 and other molecules, cellular metabolic function analysis using a Seahorse bioanalyzer and RNA sequencing analysis.

## 2. Results

### 2.1. Origin of FABP5 within the Intraocular Environment

To find the origin of FABP5 within the intraocular environment, we studied the mRNA expression of FABP5 in four types of intraocular cells, HNPCE cells, RB cells, ARPE19 cells and HOCF cells, that originate from the ciliary body, the sensory retina, the RPE and the ocular choroid, respectively. As shown in Figure 1, among these cell types, the most prominent mRNA expression of FABP5 was detected in HNPCE cells, lesser expression was also detected in ARPE 19 cells and HOCF cells, and only trace expression of FABP5 was detected in RB cells. In contrast, no expression of FABP 7 was detected in these cells (Figure 1). In addition, approximately 8.5~11.5 ng/ml of FABP5 were detected in the culture medium obtained from confluent HNPCE cells in a well of 12-well culture plate by commercially available ELISA kit for FABP5. Collectively, HNPCE cells are one of possible intraocular cells expressing FABP5 thereby secreting FABP5 into vitreous cavity.

### 2.2. Cellular Metabolic Analysis to Elucidate Possible Biological Roles of FABP5 in HNPCE Cell Line

To elucidate the possible physiological roles of FABP5 in HNPCE cells, cellular metabolic analysis by a Seahorse XFe96 Bioanalyzer was performed using FABP ligand 6, a specific inhibitor of FABP5 and FABP7. Since as stated above, mRNA expression of FABP5, but not that of FABP7, was detected in HNPCE cells, FABP ligand 6-induced effects should represent specific effects caused by suppression of the activities of FABP5 in HNPCE cells. As shown in Figure 2, treatment with FABP ligand 6 induced remarkable suppression of indices related to mitochondrial and glycolytic functions, suggesting that FABP5 indeed plays an indispensable role in survival of HNPCE cells by regulating essential cellular metabolic functions.

### 2.3. RNA Sequencing Analysis to Elucidate Underlying Mechanisms of FABP5-Related Physiological Roles in HNPCE Cell Line

To elucidate further the possible underlying mechanisms of FABP5 in HNPCE cells, RNA sequencing analysis was performed using FABP ligand 6, a specific inhibitor of FABP5 and FABP7. Since mRNA expression of FABP7 was not detected in any of the cell lines as shown in Figure 1, FABP ligand 6 should inhibit FABP5 activity in HNPCE cells. As shown in a heatmap (Figure 3) and in an M (log ratio)-A (mean average) plot (Figure 4A) and a volcano plot (Figure 4B), gene expression levels in HNPCE cells not treated with FABP ligand 6 (NT, *n* = 3) were significantly different from those in HNPCE cells treated with FABP ligand 6 (MF6, *n* = 3), and, in fact, 166 markedly up-regulated and 198 markedly down-regulated DEGs were identified between them (An excel file list of all of the DEGs is included in supplemental material and the most prominent top 10 up-regulated and down-regulated DEGs among the DEGs are shown in Table 1). 

IPA analysis was carried out to further estimate possible biological roles of FABP5 in HNPCE cells. As shown in Table 2, Table 3 and Table 4, (A) the top five canonical pathways (Table 2) were (1) cell cycle checkpoints, (2) kinetochore metaphase signaling pathway, (3) mitotic prometaphase, (4) mitotic metaphase and anaphase and (5) Rho GTP-ases activate formins, (B) the top five molecular functions (Table 3) were (1) cellular development, (2) cellular growth and proliferation, (3) cell cycle, (4) cellular assembly and organization, and (5) DNA replication, recombination and repair, and (C) the top five physiological system development and function (Table 4) were (1) organismal survival, (2) cardiovascular system development and function, (3) organismal development, (4) tissue morphology and (5) connective tissue development and function. Based on these collective observations, we speculated that FABP5 may have essential roles in cell development, cell survival and cell homeostasis. 

IPA upstream analysis was performed to estimate which molecules govern these FABP5-induced biological effects, and it was found that nuclear protein 1 (NUPR1), activating transcription factor 4 (ATF4), cyclin-dependent kinase inhibitor 1A (CDKN1A), forkhead box M1 (FOXM1), PCNA clamp associated factor (PCLAF), interleukin 6 (IL6) and amphiregulin (AREG) were upstream regulators (Table 5) and that NUPR1, hes-related family bHLH transcription factor with YRPW motif 1 (HEY1), PARP1-binding protein (PARPBP) and cytoskeleton-associated protein 2 (CKAP2) were causal network master regulators (Table 6). In support of these estimations, the results of IPA showed that (1) cellular movement, reproductive system development and function, nutritional disease, (2) cancer, hematological disease, immunological disease, (3) amino acid metabolism, small molecule biochemistry, molecular transport, (4) cell morphology, cell cycle, cellular assembly and organization, and (5) lipid metabolism, small molecule biochemistry, posttranslational modification were estimated in the top five networks (Table 7). 

### 2.4. mRNA Expression of UPR-Related Factor and Aquaporin 1 (AQP1)

To study the effects of FABP ligand 6 on unfolded protein response (UPR) of HNPCE cells, mRNA expression of major UPR-related factors including glucose-regulated protein 78 (GRP78), glucose-regulated protein 94 (GRP94), PKR-like ER kinase (PEAK), activating transcription factor 4 (ATF4), inositol-requiring enzyme-1 (IRE1) and C/EBP homologous protein (CHOP) was evaluated by qPCR. In addition, since HNPCE cells are known to be the main cells responsible for the blood-aqueous barrier (BAB) of the eye [14] and aquaporin 1 (AQP1) and Na+/K+ ATPase control the rate of aqueous humor formation and are located in the ciliary body epithelium and the iris [15,16], the effect of FABP ligand 6 on mRNA expression of AQP1 in HNPCE cells was also examined. As shown in Figure 5, the gene expression of all these UPR-related factors except GRP94 and ATF6 as well as AQP1 was significantly upregulated by FABP ligand 6.

Collectively, the results of RNA sequencing suggested that the selective FABP5 inhibitor FABP ligand 6 affects some FABP5-related signaling to regulate cellular homeostasis of UPR and BAB functions in non-adipocyte HNPCE cells. 

## 3. Discussion

The ciliary body includes three major components: the ciliary epithelium, ciliary processes, and ciliary muscle [17]. The ciliary epithelium consists of the pigmented epithelial (PE) layer and the nonpigmented epithelial (NPE) layer and the latter contributes to the production of the aqueous humor (AH) that is polarized and coupled together by gap junction proteins to form the blood-aqueous barrier (BAB) between the ciliary body stroma and the posterior chamber of the anterior segment of the eye [18,19]. NPE cells and PE cells show different gene expression profiles and thus have different phenotypic properties [20,21,22]. Phenotypic differences and functional diversities are also present among NPE cells along the distinct regions of the pars plicata, pars plana, and ora serrata of the ciliary epithelium [20,21,22]. The rate of AH secretion and the resistance to its drainage out of the eye are critical determinants to maintain levels of intraocular pressure (IOP), which fluctuate with the circadian rhythm that is affected by the light-dark cycle that is regulated in sympathetic and parasympathetic dependent manners [23], and in turn, elevated IOP is an established risk factor for the development of glaucomatous optic neuropathy [24]. Therefore, the identification of factors regulating levels of AH production by NPE cells in addition to AH drainage are extremely important factors for understanding and treating glaucomatous optic neuropathy. However, the genetic basis and molecular basis underlying both physiological and pathological conditions of the ciliary body, especially the NPE layer, have not been sufficiently investigated, although some previous studies have suggested that oxidative stress [22,25] and neuropeptides [26,27] may be importantly involved in the regulatory mechanism. In terms of FABP in the ciliary body, heart-type FABP (FABP3)-like FABP was identified in the retina and ciliary body in the chicken eye by immunohistochemistry [28]. In the present study, we first showed that the possible origin of ioFABP5 was HNPCE cells and that ioFABP5 may be critically involved in essential cellular homeostasis including mitochondrial and glycolytic functions of HNPCE cells as demonstrated by qPCR analysis (Figure 1), analysis by ELISA for FABP5, real-time cellular metabolic analysis by using a Seahorse bio-analyzer (Figure 2) and RNA sequencing analysis using IPA analysis. In support of this, it was shown that FABP5 induced an increase of proliferation, tumorigenicity and metastasis in prostate cancer, and thus an elevated level of FABP5 has been identified as a poor prognostic indicator in cancers [29,30,31]. In addition, siRNA-induced depletion of FABP5 in a prostate cancer cell line substantially inhibited tumorigenicity and proliferation in nude mice [32]. 

The mechanisms underlying the effects induced by suppression of FABP5 activity using FABP ligand 6 on mitochondrial and glycolytic functions in HNPCE cell line remain to be elucidated. However, most of the upstream regulators and causal network master regulators estimated by IPA analysis, including NUPR1 [33], ATF4 [34], CDKN1A [35], FOXM1 [36], IL6 [37], AREG) [38] and PARPBP [39], were significantly involved in the UPR, suggesting that ioFABP5 may be a critical regulator for the UPR. Analysis by qPCR showed that mRNA expression of most of UPR-related factors and AQP1 was significantly up-regulated, suggesting that FABP5 may be involved in the maintenance of biological function as a BAB in HNPCE cells. Since FABP5 has been identified as a key indicator for systemic arteriosclerosis, HT and other metabolic syndromes [40] and since ioFABP5 has been identified as a critical marker for systemic arteriosclerosis and the HT-induced retinal complication RVD [41], we speculated that the FABP5-related effect on BAB function in HNPCE cell line may also be related to systemic arteriosclerosis and HT. In fact, in our recent cohort study over a 10-year period, we found that a high level of IOP was independently associated with new onset of systemic HT [42] and this result may support our idea. 

However, as limitations of this study, the following issues need to be investigated. Firstly, the biological natures of the commercially available HNPCE cell line and other cell lines may be different from those in their in vivo native and matured conditions. Secondly, as far as we surveyed, a relationship between FABP5 and UPR has not been identified, although FABP4, which is a critical indicator for various metabolic syndromes as is FABP5 [1], has been shown to be importantly involved in UPR [43]. Thirdly, as of this writing, the contribution of FABP5 to the BAB function in HNPCE cell line is still speculative. Fourthly, several DEGs and possible important factors estimated by RNA sequencing and IPA analysis, respectively, have not been confirmed at protein levels by Western blot (WB) analysis. Fifthly, to elucidate functional roles of FABP5 in the HNPCE cell line, a Seahorse cellular metabolic measurement using a pharmacological inhibitor of FABP5 and FABP7 MF6 was only performed. Therefore, investigations to solve those unidentified issues in conjugation with additional investigation to find new key molecules among the obtained DEGs using in vitro additional functional assays, WB analysis, genetic experiments including siRNA or CRISPR/Cas knockout of the FABP5 gene, and in vivo experiments using *FABP5* deficiency mice will be our next projects.

In conclusion, FABP5-induced regulation is a critical mechanism for maintenance of intraocular homeostasis of HNPCE cell line and this may provide a clue for understanding the unidentified physiological roles of NPE. 

## 4. Materials and Methods

### 4.1. Two-Dimensional (2D) Culture of HOCF Cells, HNPCE Cells, RB Cells and ARPE19 Cells

All experimental protocols using human-derived cells were in compliance with the tenets of the Declaration of Helsinki and were approved by the internal review board of Sapporo Medical University. Human ocular choroidal fibroblast cells (HOCF cells, Cat. #6620, Science Research Laboratories, Inc., Carlsbad, CA, USA) were purchased and cultured in 2D culture dishes (150 mm) until 90% confluence at 37 °C in a recommended fibroblast medium (FM, Cat. #2301, Science Research Laboratories, Inc., Carlsbad, CA, USA). HNPCE cells (Cat. #6580, Science Research Laboratories, Inc., Carlsbad, CA, USA), RB cells (Cat. #HTB-169™, ATCC, Manassas, VA, USA) and ARPE19 cells (#CRL-2302™, ATCC, Manassas, VA, USA) were separately cultured in 150 mm planar culture dishes until they reached 90% confluence at 37 °C in a growth medium consisting of high-glucose DMEM containing 10% FBS, 1% L-glutamine, and 1% antibiotic-antimycotic. The cells were maintained by changing the medium every other day under standard humid normoxia conditions (37 °C, 5% CO_2_). All experiments described below were conducted to use 5 to 10 passaged cells.

To inhibit FABP5 activity in HNPCE cells, 10 μM of the specific FABP5 and FABP7 inhibitor FABP ligand 6 (MF6, Cat. #10010206, Cayman Chemical, Ann Abor, MI, USA) was administered.

### 4.2. Seahorse Assay

Planar cultured HNPCE cells with or without 10 μM of MF6 were subjected to analyses by a Seahorse XFe96 Bioanalyzer (Agilent Technologies) to measure the oxygen consumption rates (OCRs) and the extracellular acidification rates (ECARs) according to the manufacturer’s instructions as described previously [44]. 

Key metabolic parameters were calculated using the following formulas: Basal respiration = OCR at baseline (OCR_basal_) − OCR under rotenone/antimycin A (OCR_r/a_); ATP-linked respiration = OCR_basal_ − OCR under oligomycin (OCR_oligo_); Proton leak = OCR_oligo_ − OCRr/a; Maximal respiration = OCR under FCCP (OCR_FCCP_) − OCR_r/a_; Non-mitochondrial respiration = OCR_r/a_; Basal ECAR = ECAR at baseline (ECAR_basal_) − ECAR at the last measurement under 2-DG (ECAR_2-DG_); Glycolytic Capacity = ECAR under oligomycin (ECAR_oligo_) − ECAR_2-DG_; Non-glycolytic acidification = ECAR_2-DG_; Baseline OCR/ECAR = OCR_basal_/ECAR_basal_.

### 4.3. RNA Sequencing Analysis of Gene Functions and Analysis of Pathways

Total RNA was obtained from 2D confluent cells of HNPCE cell line that were untreated or treated with 10 μM FABP ligand 6 for 24 h in a 150 mm dish using an RNeasy mini kit (Qiagen, Valencia, CA, USA) and then RNA extraction and next-generation sequencing were performed as described recently [45]. The sequence data obtained were filtered using FastQC software (version 0.11.7), and their quality was checked by an Agilent 2100 Bioanalyzer (Agilent, CA, USA) and Trimmomatic (version 0.38) and then they were mapped to the reference genome sequence (GRCh38) using HISAT2 tools software (HISAT 2.2.1, accessed on 1 June 2024) [46]. The read counting for each respective gene and statistical analysis were processed using featureCounts (version 1.6.3) and DESeq2 (version 1.24.0), respectively. Differentially expressed genes (DEGs) were determined as genes with fold-change ≧2.0 and false discovery rate (FDR)-adjusted *p*-value <  0.05 and *q*  <  0.08 between groups. 

Ingenuity pathway analysis (IPA, Qiagen, https://www.qiagenbioinformatics.com/products/ingenuity-pathway-analysis, accessed on 1 June 2024) [47] was used for further analysis to predict various pathways by uploading an excel file of the significantly up-regulated and down-regulated DEGs to IPA core analyses. Enrichment of the particular genes in networks in IPA was evaluated using Fisher’s exact test. In addition, the IPA software (IPA, Qiagen, https://www.qiagenbioinformatics.com/products/ingenuity-pathway-analysis, accessed on 1 June 2024) orders top functions related to each network based on the enrichment scores (z-score) and predicts possible upstream regulators and causal network regulators as shown in recent studies [47,48,49].

### 4.4. Other Analytical Methods

Total RNA was extracted from the various planar cultured cells as described above, and reverse transcription and quantitative real-time PCR (qRT-PCR) were performed using specific primers and probes (Appendix A) as previously reported [50]. In brief, total RNA was extracted using an RNeasy mini kit (QIAGEN, Valencia, CA, USA) according to the manufacturer’s instructions. Reverse transcription was performed by using the SuperScript IV kit (Invitrogen, Waltham, MA, USA) according to the manufacturer’s protocols. Then qRT-PCR was performed with Universal Taqman Master mix by using a StepOnePlus system (Applied Biosystems/Thermo Fisher Scientific, Waltham, MA, USA). Each gene expression level was normalized to the expression of internal control 36B4 (Rplp0). 

As experimental data, arithmetic means ± standard errors of the mean (SEMs) were used in conjugation with statistical analyses essentially as described in our previous report [50]. A significant difference less than 0.05 between two groups was determined by Student’s *t*-test and that among matched multiple group comparisons, one-way ANOVA followed by a Tukey’s multiple comparison test using Graph Pad Prism 8 software (GraphPad Software, San Diego, CA, USA) as described in our recent reports [50,51]. 

## Figures and Tables

**Figure 1 ijms-25-09285-f001:**
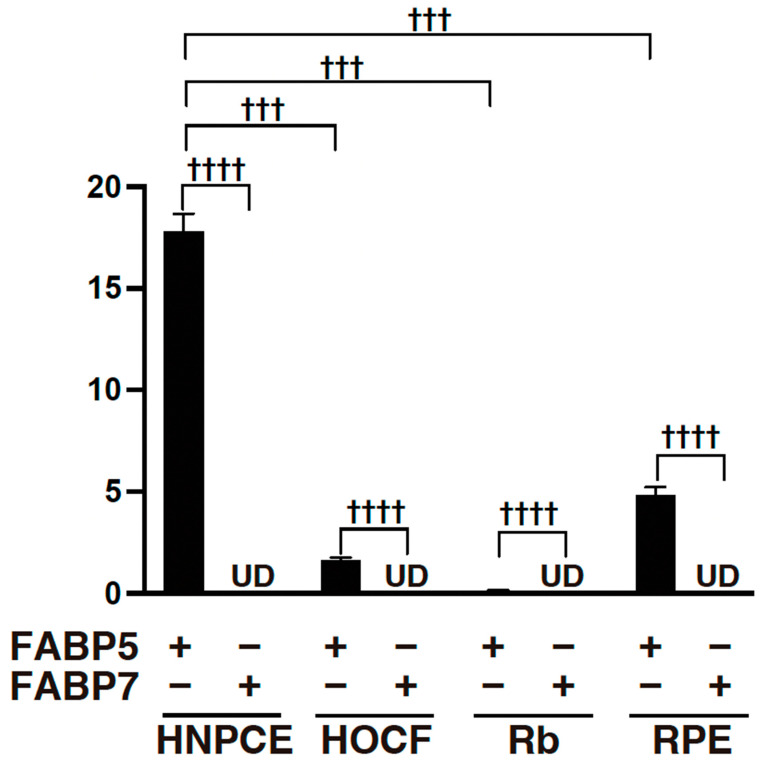
mRNA expression of FABP5 and FABP7 in HNPCE, RB, ARPE19 and HOCF cells. 2D cultured HNPCE, Rb, ARPE19 and HOCF cells were subjected to qPCR analysis and the mRNA expression of FABP5 and FABP7 was estimated. Experiments were repeated three times using freshly prepared cells (*n* = 3 each) in each experiment. ^††††^ *p* < 0.001 (student *t*-test). ^†††^ *p* < 0.005 (one-way ANOVA followed by a Tukey’s multiple comparison test). UD: under detectable levels.

**Figure 2 ijms-25-09285-f002:**
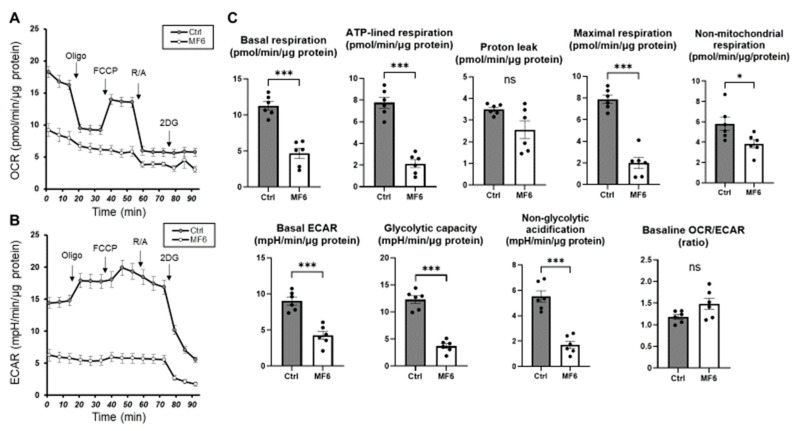
Effects of FABP ligand 6 on cellular metabolic functions. Planar 2D cultured HNPCE cells were not treated (Ctrl) or were treated with 10 μM FABP ligand 6 (MF6), a specific inhibitor of FABP5 and FABP7, and each sample (*n* = 6) was loaded to real-time metabolic function analysis using a Seahorse XFe96 Bioanalyzer. (**A**) Measurements of oxygen consumption rate (OCR). (**B**) Measurements of extracellular acidification rate (ECAR). (**C**) Key indices of metabolic parameters. ns: non-significant statistics, * *p* < 0.05, and *** *p* < 0.001 (student *t*-test). Oligo: oligomycin. FCCP: carbonyl cyanide p-trifluoromethoxyphenylhydrazone. R/A: a mixture of rotenone/antimycin A 2DG: 2-deoxyglucose.

**Figure 3 ijms-25-09285-f003:**
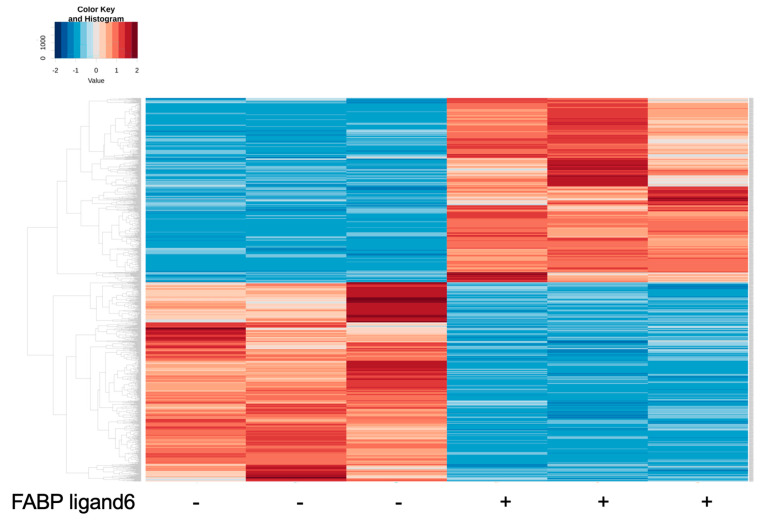
Heatmap for DEGs in HNPCE cells not treated with FABP ligand 6 (NT 1–3) and HNPCE cells treated with FABP ligand 6 (MF 1-3). 2D cultured HNPCE cells not treated with FABP ligand 6 (NT, *n* = 3) and those treated with 10 μM of FABP ligand 6 (MF6 *n* = 3) were loaded to RNA sequencing analysis. Differentially expressed genes (DEGs) are shown by a hierarchical clustering heatmap. Colored bars represent either overexpressed (red) or underexpressed (blue) DEGs in NT cells compared with those in MF6 cells.

**Figure 4 ijms-25-09285-f004:**
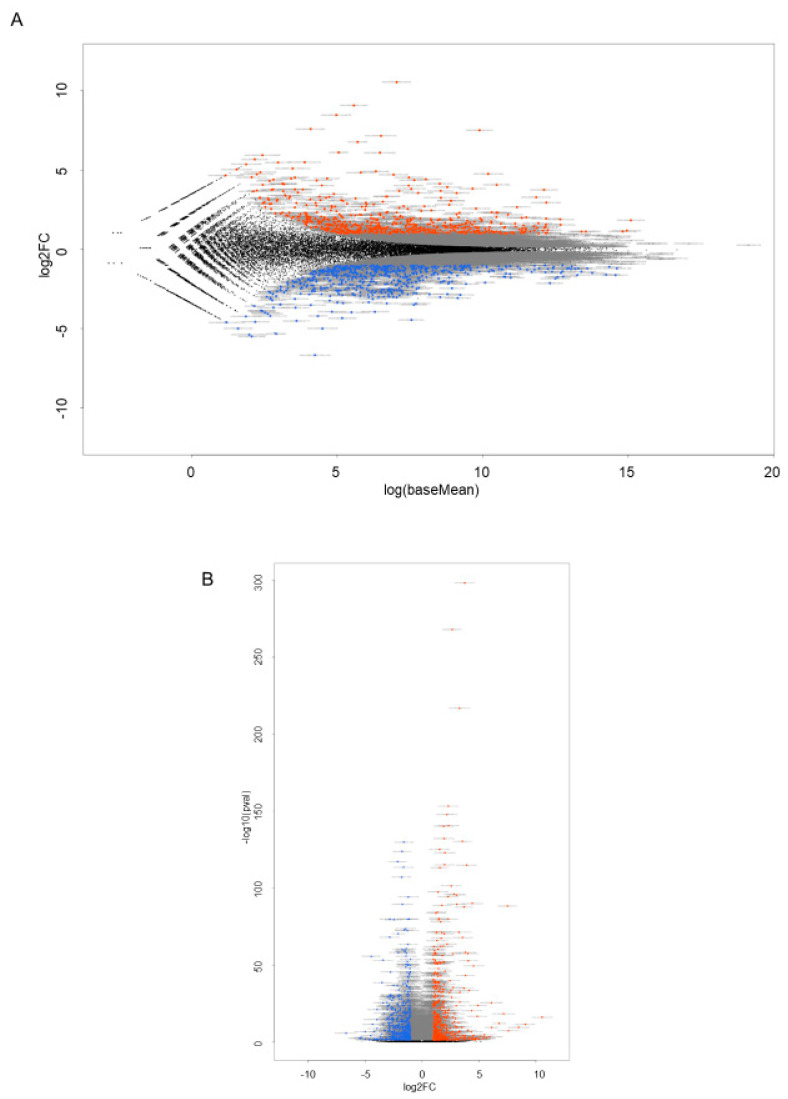
M (log ratio)–A (mean average) plot (**A**) and volcano plot (**B**) in HNPCE cells not treated with FABP ligand 6 and HNPCE cells treated with FABP ligand 6. Differentially expressed genes (DEGs) are shown by an M–A plot (**A**) and a volcano plot (**B**). Colored points represent either overexpressed (red) or underexpressed (blue) DEGs in NT cells (*n* = 3) compared with those in MF6 cells (*n* = 3).

**Figure 5 ijms-25-09285-f005:**
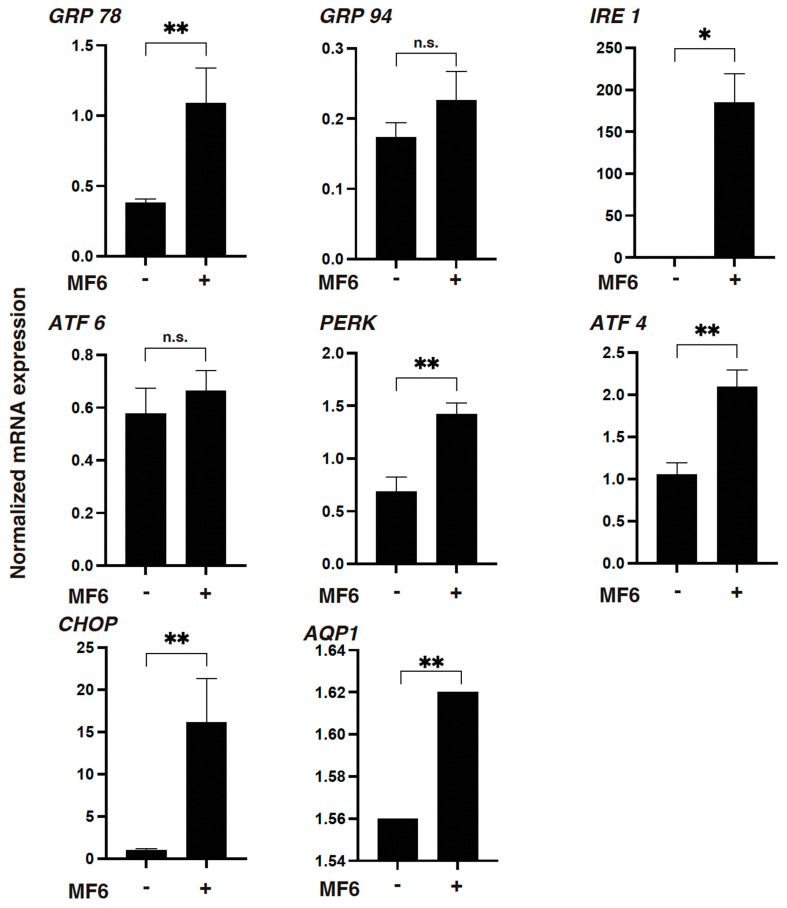
qPCR analysis for UPR-related factors and AQP1. 2D cultured HNPCE cells were subjected to qPCR analysis, and the mRNA expression of UPR–related factors, including *GRP78*, *GRP94*, *PEAK*, *ATF4*, *IRE* and *CHOP*, and *AQP1* was estimated. Experiments were repeated three times using freshly prepared cells (*n* = 3 each) in each experiment. n.s.: non-significant statistics, * *p* < 0.05 and ** *p* < 0.01 (student *t*-test).

**Table 1 ijms-25-09285-t001:** Top 10 up-regulated and down-regulated DEGs between FABP ligand 6-untreated HNPCE cells and FABP ligand 6-treated HNPCE cells.

Up-Regulation		Down-Regulation	
Molecules	*p*-Value	Molecules	*p*-Value
DUSP15	10.544	SCEL-AS1	−6.667
LAMP3	9.076	INHBB	−5.489
KLHDC7B	8.471	LOC124902532	−5.380
SLC30A2	7.591	AC1230231	−5.323
NGFR	7.513	FAM111B	−4.993
NGFR-AS1	7.156	IGLON5	−4.992
RP11_81K22	6.769	RP5_1028K72	−4.602
CTD_3118D112	6.112	ANXA8/ANXA8L1	−4.571
ATP6V0D2	6.097	MT-TT	−4.505
LOC105369568	5.942	SCEL	−4.448

**Table 2 ijms-25-09285-t002:** Top 5 molecular and cellular functions.

Name	*p*-Value
Cell Cycle Checkpoints	3.15 × 10^−23^
Kinetochore Metaphase Signaling Pathway	1.32 × 10^−21^
Mitotic Prometaphase	9.90 × 10^−21^
Mitotic Metaphase and Anaphase	1.63 × 10^−20^
RHO GTPase Activate Formins	6.27 × 10^−20^

**Table 3 ijms-25-09285-t003:** Top 5 canonical pathways.

Name	*p*-Value Range
Cellular Development	3.00 × 10^−6^–2.31 × 10^−30^
Cellular Growth and Proliferation	3.00 × 10^−6^–2.31 × 10^−30^
Cell Cycle	2.58 × 10^−6^–5.10 × 10^−30^
Cellular Assembly and Organization	2.79 × 10^−6^–9.10 × 10^−29^
DNA Replication, Recombination, and Repair	2.38 × 10^−6^–9.10 × 10^−29^

**Table 4 ijms-25-09285-t004:** Top 5 networks.

Name	*p*-Value Range
Organismal Survival	2.71 × 10^−7^–7.48 × 10^−37^
Cardiovascular System Development and Function	2.60 × 10^−6^–4.03 × 10^−21^
Organismal Development	3.00 × 10^−6^–4.03 × 10^−21^
Tissue Morphology	2.60 × 10^−6^–4.44 × 10^−20^
Connective Tissue Development and Function	1.68 × 10^−6^–2.28 × 10^−13^

**Table 5 ijms-25-09285-t005:** Upstream regulator.

Upstream Regulator	Expr Log/Ratio	Molecule Type	Activation z-Score
NUPR1	4.115	transcription regulator	6.199
ATF4	1.570	transcription regulator	5.362
CDKN1A	1.551	kinase	3.367
FOXM1	−1.823	transcription regulator	−5.520
PCLAF	−2.458	other	−4.266
IL6	1.557	cytokine	1.576
AREG	1.215	growth factor	−2.557

**Table 6 ijms-25-09285-t006:** Causal Networks.

Master Regulator	Expr Log/Ratio	Molecule Type	Activation z-Score
NUPR1	4.115	transcription regulator	6.353
HEY1	1.093	transcription regulator	4.932
PARPBP	−1.108	other	−5.647
CKAP2L	−2.625	other	−6.245

**Table 7 ijms-25-09285-t007:** Top Networks.

Associated Network Functions	Score
Cellular, Movement, Reproductive System Development and Function, Nutritional Disease	46
Cancer, Hematological Disease, Immunological Disease	39
Amino Acid Metabolism, Small Molecule Biochemistry, Molecular Transeport	39
Cell Morphology, Cell Cycle, Cellular Assembly and Organization	37
Lipid Metabolism, Small Molecule Biochemistry, Post-Translational Modification	34

## Data Availability

The data can be shared upon request.

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
