# Peer review of "FABP5 Is a Possible Factor for the Maintenance of Functions of Human Non-Pigmented Ciliary Epithelium Cells"

_ijms, 2024, doi:10.3390/ijms25179285_

Round 1

Reviewer 1 Report

Comments and Suggestions for Authors

Title: FABP5 is a crucial factor for the maintenance of functions of human non-pigmented ciliary epithelium cells as a blood aqueous barrier.

After reading through this manuscript It is unclear why the authors believe that FABP5 necessarily has any involvement (direct or indirect) in the maintenance of the blood aqueous barrier (BAB).  Please justify.

Abstract:

The abbreviation HNPCE appears in places as HNCPE (line 20 and line 25).  Please pick a consistent style and use it throughout. 

Once again, please justify the final sentence which suggests that FABP5 may be involved in maintenance of the BAB.  And please explain the apparent discrepancy between ‘FABP5 may be involved in maintenance of the BAB’ (as stated in the Abstract) and ‘FABP5 is a crucial factor in maintenance of the BAB’ (as stated in the article title).

Introduction (Line 53).  Please indicate the reason(s) why vitreous was removed from patients with non-retinal vascular diseases (RVD).  

Figure 3.  Please define the abbreviation, M-A plot.

Methods:  Did the authors not consider relating mRNA expression to corresponding protein expression via Western blotting?  Without this it is not possible to be certain that upregulation of gene transcription actually leads to upregulation of gene translation.  Please offer some comment. 

Discussion:  line 266.  The authors state that the contribution to FABP5 to BAB function in HNPCE cells is still speculative.  How does this align with prior statements in relation to FABP5 vis-à-vis the BAB as made in the Manuscript Title and the Abstract?

The English is good, but there are a few minor grammatical errors:

Introduction (line 58): I suggest that, ‘were not merely originated from peripheral blood’ is changed to to ‘had not merely originated from peripheral blood’.

Introduction (line 65): I suggest that, ‘we first investigated to elucidate’ is changed to ‘we first aimed to elucidate’.

Results (line 155).  I suggest that, ‘Since as shown as above, mRNA’ is changed to ‘since as shown above, mRNA’.

Legend to Figure 5 (line 164).  Please close gap in 10    M FABP ligand, so that it reads, ‘10M FABP ligand’.  Also, was the concentration really 10M, or is the text missing a special character here?

Results (line 191).  Please change ‘unfolded protein responce’ to ‘unfolded protein response’.

Discussion (line 221).  Intraocular pressure (IOP) is determined by various factors, including: the rate of secretion of aqueous humor, and the resistance to its drainage (not just its drainage) out of the eye.  

Discussion (line 239).  I suggest that, ‘In support this’ is changed to ‘In support of this’.

Discussion (line 250).  I suggest that, ‘mRNA expression of the most of UPR-related factors’ is changed to, ‘mRNA expression of most of the UPR-related factors’.

Comments on the Quality of English Language

The quality of the written English is high, however minor corrections to a few grammatical errors are required, as indicated in the Comments to Authors section.

Author Response

Title: FABP5 is a crucial factor for the maintenance of functions of human non-pigmented ciliary epithelium cells as a blood aqueous barrier.

  1. After reading through this manuscript It is unclear why the authors believe that FABP5 necessarily has any involvement (direct or indirect) in the maintenance of the blood aqueous barrier (BAB). Please justify.

Answer; We sincerely appreciate your excellent comment. I totally agree with your comment that our statement of the blood aqueous barrier (BAB) is overstatement. Therefore, words ‘as a blood aqueous barrier ‘ in the title and abstract are removed.

Abstract:

  1. The abbreviation HNPCE appears in places as HNCPE (line 20 and line 25). Please pick a consistent style and use it throughout.

Answer; We sincerely appreciate your excellent comment. As pointed out, The abbreviation HNPCE was unified throughout manuscript.

  1. Once again, please justify the final sentence which suggests that FABP5 may be involved in maintenance of the BAB. And please explain the apparent discrepancy between ‘FABP5 may be involved in maintenance of the BAB’ (as stated in the Abstract) and ‘FABP5 is a crucial factor in maintenance of the BAB’ (as stated in the article title).

Answer; We sincerely appreciate your excellent comment. Again, I totally agree with your comment that our statement of the blood aqueous barrier (BAB) is overstatement. Therefore, words ‘as a blood aqueous barrier ‘ in the title and abstract are removed.

  1. Introduction (Line 53). Please indicate the reason(s) why vitreous was removed from patients with non-retinal vascular diseases (RVD). 

Answer; We sincerely appreciate your excellent comment. In terms of non-RVD, patients with epiretinal membrane were included and thus this information is included: “the levels in vitreous fluid obtained from patients with epiretinal membrane (non-RVD) [12,13]”.

  1. Figure 3. Please define the abbreviation, M-A plot.

Answer; We sincerely appreciate your excellent comment. As suggested, it is defined “M (log ratio)-A (mean average) plot”.

  1. Methods: Did the authors not consider relating mRNA expression to corresponding protein expression via Western blotting?  Without this it is not possible to be certain that upregulation of gene transcription actually leads to upregulation of gene translation.  Please offer some comment.

Answer; We sincerely appreciate your excellent comment. I agree that additional data by WB analysis will be necessary to confirm changes in mRNA expression, and therefore, this information is included in the study limitation in the discussion: ‘However, as limitations of this study, the following issues need to be investigated. Firstly, the biological natures of the commercially available HNPCE cells and other cells may be different from those in their in vivo native and matured conditions. Secondly, as far as we surveyed, a relationship between FABP5 and UPR has not been identified, although FABP4, which is a critical indicator for various metabolic syndromes as is FABP5 [1], has been shown to be importantly involved in UPR [43]. Thirdly, as of this writing, the contribution of FABP5 to the BAB function in HNPCE cells is still speculative. Fourthly, several DEGs and possible important factors estimated by RNA sequencing and IPA analysis, respectively have not been confirmed at protein levels by western blot (WB) analysis. Therefore, investigations to solve those unidentified issues in conjugation with additional investigation to find new key molecules among the obtained DEGs using in vitro additional functional assays, WB analysis and in vivo experiments using FABP5 deficiency mice will be our next projects.’.

  1. Discussion: line 266.  The authors state that the contribution to FABP5 to BAB function in HNPCE cells is still speculative.  How does this align with prior statements in relation to FABP5 vis-à-vis the BAB as made in the Manuscript Title and the Abstract?

Answer; We sincerely appreciate your excellent comment. I totally agree with your comment that our statement of the blood aqueous barrier (BAB) is overstatement. Therefore, words ‘as a blood aqueous barrier ‘ in the title and abstract are removed

  1. The English is good, but there are a few minor grammatical errors: Introduction (line 58): I suggest that, ‘were not merely originated from peripheral blood’ is changed to to ‘had not merely originated from peripheral blood’.

Answer; We sincerely appreciate your excellent comment. As pointed out, that is changed to ‘had not merely originated from peripheral blood’.

  1. Introduction (line 65): I suggest that, ‘we first investigated to elucidate’ is changed to ‘we first aimed to elucidate’.

Answer; We sincerely appreciate your excellent comment. As pointed out, that is changed to ‘we first aimed to elucidate’.

  1. Results (line 155). I suggest that, ‘Since as shown as above, mRNA’ is changed to ‘since as shown above, mRNA’.

Answer; We sincerely appreciate your excellent comment. As pointed out, that is changed to ‘Since as shown above, mRNA’.

  1. Legend to Figure 5 (line 164). Please close gap in 10    M FABP ligand, so that it reads, ‘10M FABP ligand’.  Also, was the concentration really 10M, or is the text missing a special character here?

Answer; We sincerely appreciate your excellent comment. As pointed out, this is a careless mistake and 10 mM is correct. Therefore this is fixed.

  1. Results (line 191). Please change ‘unfolded protein responce’ to ‘unfolded protein response’.

Answer; We sincerely appreciate your excellent comment. As pointed out this I a careless mistake and therefore, this is fixed.

  1. Discussion (line 221). Intraocular pressure (IOP) is determined by various factors, including: the rate of secretion of aqueous humor, and the resistance to its drainage (not just its drainage) out of the eye. 

Answer; We sincerely appreciate your excellent comment. As pointed out, ‘its drainage’ is changed to ‘the resistance to its drainage’.

  1. Discussion (line 239). I suggest that, ‘In support this’ is changed to ‘In support of this’.

Answer; We sincerely appreciate your excellent comment. As pointed out, this is changed to ‘In support of this’.

  1. Discussion (line 250). I suggest that, ‘mRNA expression of the most of UPR-related factors’ is changed to, ‘mRNA expression of most of the UPR-related factors’.

Answer; We sincerely appreciate your excellent comment. As pointed out, this is changed to ‘mRNA expression of most of the UPR-related factors’.

Comments on the Quality of English Language

  1. The quality of the written English is high, however minor corrections to a few grammatical errors are required, as indicated in the Comments to Authors section.

Answer; We sincerely appreciate your excellent comment. Quality of English is carefully checked by a native English-speaking scientist and his certificate is attached.

Reviewer 2 Report

Comments and Suggestions for Authors

Comments on the review manuscript

FABP5 is a crucial factor for the maintenance of functions of human non-pigmented ciliary epithelium cells as a  blood-aqueous barrier 

Higashide et al IJMS 2024

In this manuscript, the authors investigated the role of FABP5 in intraocular physiology using various ocular cell lines. Employing FABP ligand 6, a specific inhibitor of FABP5 and FABP7, the authors conducted qPCR analysis for FABP5, FABP7, and other molecules, Seahorse-based cellular metabolic function analysis, and RNA sequencing. Authors concluded that their findings of this study suggest that FABP5 originates from the ciliary body and may be crucial for maintaining UPR and AQP1-related functions of HNPCE cells, which serve as a blood-aqueous barrier

Broad comments

The role of FABP5 in intraocular pathologies is not well-explored, which adds novelty and importance to this manuscript. Previous studies by the authors have established a direct link between VEGF and FABP5, emphasizing its relevance in retinal vasculopathies. FABP5 could be considered as a potential therapeutic target for retinal vascular diseases (RVD). However, the current data may lead to overinterpretation of its role. For example, the role of FABP5 in the maintenance of the blood-aqueous barrier (BAB) is not clearly demonstrated in this study. Therefore, the authors are encouraged to revise the title and focus more on the role of FABP5 in maintaining non-pigmented ciliary epithelium cells, both in the title and throughout the abstract and discussion.

The current manuscript has several broad issues as follows:

1.        The manuscript would benefit from further language refinement to enhance clarity and readability.

  1. The authors are advised to follow the IJMS journal style.
  2. The Results section should be written with subpoints and subheadings to enhance readability and data comprehension.
  3. The result figures are not properly presented, often incomplete without proper labeling, statistics, or error bars. The figure legends can also be improved.
  4. The representation style of figures can be further refined for better understanding of the results.
  5. The Methods section needs more details. It should also include a ‘Statistics’ section highlighting the methods and tools used.

Other technical/general queries and suggestions:

·      Figure 1. mRNA expression of FABP5 and FABP7 in HNPCE, RB, ARPE19 and HOCF cells. 2D 92 cultured HNPCE, Rb, ARPE19 and HOCF cells were subjected to qPCR analysis and the mRNA 93 expression of FABP5 and FABP7 was estimated. Experiments were repeated three times using 94 freshly prepared cells (n=3 each) in each experiment. ***P<0.005. “

For better understanding, simplify the figure. Using two separate color bars to represent FABP5 and FABP7 might be more effective than using ‘+-‘ signs. Ensure to include the P value or * on the bar diagrams. Additionally, mention the passage number of the cells used for qPCR in the Methods section.

·      “To inhibit FABP5 activity in HNCPE cells, 10 M of the specific FABP5 and FABP7 286 inhibitor FABP ligand 6 (MF6, Cat. #10010206, Cayman Chemical, Ann Abor, MI USA) 287 was administered. 

Please correct the error of FABP Ligand 6 concentration: it should be 10 µM, not 10 M

·      Other analytical methods 322 Real-time PCR was carried out essentially as previously reported [26,27] using pre- 323 designed primers (supplemental Table 1). The expression of each respective gene was nor- 324 malized by using the expression of the housekeeping gene 36B4 (Rplp0). As experimental 325 data, the arithmetic mean ± standard error of the mean (SEM) was used in conjugation 326 with statistical analyses essentially as described in our previous reports [26,27]. 327 (MF6, Cat. #10010206, Cayman Chemical, Ann Abor, MI USA) 287 was administered. 

Elaborate the qPCR protocol in detail. Highlight the fold change calculation formula (Normalized mRNA expression??)

Figure 3. M-A plot (A) and volcano plot (B) for HNPCE cells not treated with FABP ligand 6-un- 122 treated HNCPE cells (NT 1-3) and FABP ligand 6-treated HNCPE cells (MF 1-3). 2D cultured 123 HNCPE cells not treated with FABP ligand 6 (NT, n=3) and those treated with 10 μM of FABP ligand 124 6 (MF6 n=3) were loaded to RNA sequencing analysis. Differentially expressed genes (DEGs) are 125 shown by an M–A plot (A) and a volcano plot (B). Colored points represent either overexpressed 126 (red) or underexpressed (blue) DEGs in NT cells compared with those in MF6 cells. 12 

Simplify the legend and remove repetitive statements.

·      “Figure 4. Results of GO enrichment analysis. (A) Biological process, (B) Cell component and (C) 133 Molecular function. Bar color represents p values and x-axis represents numbers of DEGs. “

Where are subfigures 'B' and 'C' in Figure 4? Please rectify the figure legend

·      “Figure 5. Effects of FABP ligand 6 on cellular metabolic functions. Planar 2D cultured HNCPE 163 cells were not treated (Ctrl) or were treated with 10 M FABP ligand 6 (MF6), a specific inhibitor of 164 FABP5 and FABP7, and each sample (n=6) was loaded to real-time metabolic function analysis using 165 a Seahorse XFe96 Bioanalyzer. A) Measurements of oxygen consumption rate (OCR). B) Measure- 166 ments of extracellular acidification rate (ECAR). C) Key indices of metabolic parameters. *P<0.05, 167 **P<0.01, and ***P<0.001. Oligo: oligomycin. FCCP: carbonyl cyanide p-trifluoromethoxyphenylhy- 168 drazone. R/A: a mixture of rotenone/antimycin A 2DG: 2-deoxyglucose”

Highlight non-significant statistics as 'ns' on the graph figures.

·      “Figure 6. qPCR analysis for UPR-related factors and AQP1. 2D cultured HNPCE were subjected 206 to qPCR analysis, and the mRNA expression of UPR-related factors, including GRP78, GRP94, 207 PEAK, ATF4, IRE and CHOP, and AQP1 was estimated. Experiments were repeated three times 208 using freshly prepared cells (n=3 each) in each experiment. *P<0.05, **P<0.01. “

Correct the figure graphs as follows:

§  For AQP1: Label each bar, error bar, and statistics.

§  For GRP78, GRP94, and IRE1: Label each individual bar, error bar, and statistics. 

§  Highlight non-significant results as 'NS'

“To study the effects of FABP ligand 6 on unfolded protein responce (UPR) of HNPCE 191 cells, mRNA expression of major UPR-related factors including glucose-regulated protein 192 78 (GRP78), glucose-regulated protein 94 (GRP94), PKR-like ER kinase (PEAK), activating 193 transcription factor 4 (ATF4), inositol requiring enzyme-1 (IRE1) and C/EBP homologous 194 protein (CHOP) was evaluated by qPCR. In addition, since HNPCE cells are known as the 195 main cell responsible for the blood-aqueous barrier (BAB) of the eye [28] and aquaporin 1 196 (AQP1) and Na+/K+ ATPase control the rate of aqueous humor formation and are located 197 in the ciliary body epitheliums and the iris [29,30], the effect of FABP ligand 6 on mRNA 198 expression of AQP1 in HNPCE cells was also studied. As shown in Fig. 6, gene expression 199 of all of these UPR-related factors and AQP1 were significantly upregulated by FABP lig- 200 and 6. “

Please present the results with appropriate subheadings. The results section should focus on data representation, while literature references can be included in the discussion section. Highlight the results clearly, including exact fold changes in expression compared to the control and corresponding p-values. For example, in the AQP1 qPCR bar graph, significance is not highlighted, and the axis labels are not properly formatted. Additionally, the AQP1 expression between the control and MF6 group does not appear to show a difference (MF6/Control ~1.04)

Comments on the Quality of English Language

The manuscript would benefit from further language refinement to enhance clarity and readability.

Author Response

FABP5 is a crucial factor for the maintenance of functions of human non-pigmented ciliary epithelium cells as a blood-aqueous barrier

Higashide et al IJMS 2024

In this manuscript, the authors investigated the role of FABP5 in intraocular physiology using various ocular cell lines. Employing FABP ligand 6, a specific inhibitor of FABP5 and FABP7, the authors conducted qPCR analysis for FABP5, FABP7, and other molecules, Seahorse-based cellular metabolic function analysis, and RNA sequencing. Authors concluded that their findings of this study suggest that FABP5 originates from the ciliary body and may be crucial for maintaining UPR and AQP1-related functions of HNPCE cells, which serve as a blood-aqueous barrier

  1. Broad comments: The role of FABP5 in intraocular pathologies is not well-explored, which adds novelty and importance to this manuscript. Previous studies by the authors have established a direct link between VEGF and FABP5, emphasizing its relevance in retinal vasculopathies. FABP5 could be considered as a potential therapeutic target for retinal vascular diseases (RVD). However, the current data may lead to overinterpretation of its role. For example, the role of FABP5 in the maintenance of the blood-aqueous barrier (BAB) is not clearly demonstrated in this study. Therefore, the authors are encouraged to revise the title and focus more on the role of FABP5 in maintaining non-pigmented ciliary epithelium cells, both in the title and throughout the abstract and discussion.

Answer; We sincerely appreciate your excellent comment. I totally agree with your comment that our statement of the blood aqueous barrier (BAB) is overstatement. Therefore, words ‘as a blood aqueous barrier‘ in the title and abstract are removed. In addition, last sentence of 2nd paragraph of Discussion was changed to ‘In fact, in our recent cohort study over a 10-year period, we found that a high level of IOP was independently associated with new onset of systemic HT [42] and this result may support our idea.’.

The current manuscript has several broad issues as follows:

  1. The manuscript would benefit from further language refinement to enhance clarity and readability.

Answer; We sincerely appreciate your excellent comment. As suggested, quality of English is again checked by a native English-speaking scientist.

  1. The authors are advised to follow the IJMS journal style.

Answer; We sincerely appreciate your excellent comment. As suggested, I again checked to follow IJMS journal style.

  1. The Results section should be written with subpoints and subheadings to enhance readability and data comprehension.

Answer; We sincerely appreciate your excellent comment. As suggested, orders of paragraph are changed: 1) origin of FABP5, 2) cellular metabolic analysis, 3) RNA sequencing and 4) qPCR, and subheadings are also included.

  1. The result figures are not properly presented, often incomplete without proper labeling, statistics, or error bars. The figure legends can also be improved. The representation style of figures can be further refined for better understanding of the results.

Answer; We sincerely appreciate your excellent comment. As pointed out with other comments (#7, #12, #13), we carefully revised these figures.

  1. The Methods section needs more details. It should also include a ‘Statistics’ section highlighting the methods and tools used.

Answer; We sincerely appreciate your excellent comment. As pointed out, statistical analysis was done by student-t-test and one-way ANOVA followed by a Tukey's multiple comparison test for comparisons between two groups and among more than three groups, and therefore, this information is included: ‘A significant difference less than 0.05 between two groups was determined by student t-test and that among matched multiple group comparisons, one-way ANOVA followed by a Tukey's multiple comparison test using Graph Pad Prism 8 software (GraphPad Software, San Diego, CA) as described in our recent reports [50,51].’.

Other technical/general queries and suggestions:

  1. “Figure 1. mRNA expression of FABP5 and FABP7 in HNPCE, RB, ARPE19 and HOCF cells. 2D 92 cultured HNPCE, Rb, ARPE19 and HOCF cells were subjected to qPCR analysis and the mRNA 93 expression of FABP5 and FABP7 was estimated. Experiments were repeated three times using 94 freshly prepared cells (n=3 each) in each experiment. ***P<0.005. “ For better understanding, simplify the figure. Using two separate color bars to represent FABP5 and FABP7 might be more effective than using ‘+-‘ signs. Ensure to include the P value or * on the bar diagrams. Additionally, mention the passage number of the cells used for qPCR in the Methods section.

Answer; We sincerely appreciate your excellent comment. As suggested, I agree with difficulty in distinguish two bars representing FABP5 and FABP7. Nevertheless, since expression levels of FABP7 of four different cells were all under detectable levels, it seems difficult to show this bar with different color with FABP5. Therefore, UD: under detectable levels was included on the FABP7. In addition, statistical difference between FABP5 and FABP7 in each different cell and that among groups were indicated in figure 1 and legend: ‘2D cultured HNPCE, Rb, ARPE19 and HOCF cells were subjected to qPCR analysis and the mRNA expression of FABP5 was estimated. Experiments were repeated three times using freshly prepared cells (n=3 each) in each experiment. ***P<0.005 and ****P<0.001 (student t-test), †††P<0.005 and ††††P<0.001, UD: under detectable levels.’. In terms of cell passaging, we used 5 to 10 passaged cells for all experiments, and therefore this information is included in the method: ‘All experiments described below were conducted to use 5 to 10 passaged cells.’.

  1. “To inhibit FABP5 activity in HNCPE cells, 10 mM of the specific FABP5 and FABP7 286 inhibitor FABP ligand 6 (MF6, Cat. #10010206, Cayman Chemical, Ann Abor, MI USA) 287 was administered. “Please correct the error of FABP Ligand 6 concentration: it should be 10 µM, not 10 M

Answer; We sincerely appreciate your excellent comment. As pointed out, this is a careless mistake and 10 mM is correct. Therefore, this is fixed.

  1. “Other analytical methods 322 Real-time PCR was carried out essentially as previously reported [26,27] using pre- 323 designed primers (supplemental Table 1). The expression of each respective gene was nor- 324 malized by using the expression of the housekeeping gene 36B4 (Rplp0). As experimental 325 data, the arithmetic mean ± standard error of the mean (SEM) was used in conjugation 326 with statistical analyses essentially as described in our previous reports [26,27]. 327 (MF6, Cat. #10010206, Cayman Chemical, Ann Abor, MI USA) 287 was administered. “Elaborate the qPCR protocol in detail. Highlight the fold change calculation formula (Normalized mRNA expression??)

Answer; We sincerely appreciate your excellent comment. As suggested, more detail of qPCR procedures are included: ‘Total RNA was extracted from the various planar cultured cells as described above, and reverse transcription and quantitative real-time PCR (qRT-PCR) were performed using specific primers and probes (Supplementary Table S1) as previously reported [50]. In brief, total RNA was extracted using an RNeasy mini kit (QIAGEN, Valencia, CA, USA) according to the manufacturer’s instructions. Reverse transcription was performed by using the SuperScript IV kit (Invitrogen) according to the manufacturer’s protocols. Then qRT-PCR was performed with Universal Taqman Master mix by using a StepOnePlus system (Applied Biosystems/Thermo Fisher Scientific). Each gene expression level was normalized to the expression of internal control 36B4 (Rplp0).’.

  1. “Figure 3. M-A plot (A) and volcano plot (B) for HNPCE cells not treated with FABP ligand 6-un- 122 treated HNCPE cells (NT 1-3) and FABP ligand 6-treated HNCPE cells (MF 1-3). 2D cultured 123 HNCPE cells not treated with FABP ligand 6 (NT, n=3) and those treated with 10 μM of FABP ligand 124 6 (MF6 n=3) were loaded to RNA sequencing analysis. Differentially expressed genes (DEGs) are 125 shown by an M–A plot (A) and a volcano plot (B). Colored points represent either overexpressed 126 (red) or underexpressed (blue) DEGs in NT cells compared with those in MF6 cells. 12 “Simplify the legend and remove repetitive statements.

Answer; We sincerely appreciate your excellent comment. As suggested, this legend is simplified to avoid repetition to prior legend: ‘Differentially expressed genes (DEGs) are shown by an M–A plot (A) and a volcano plot (B). Colored points represent either overexpressed (red) or underexpressed (blue) DEGs in NT cells (n=3) compared with those in MF6 cells (n=3).’

  1. “Figure 4. Results of GO enrichment analysis. (A) Biological process, (B) Cell component and (C) 133 Molecular function. Bar color represents p values and x-axis represents numbers of DEGs. “Where are subfigures 'B' and 'C' in Figure 4? Please rectify the figure legend

Answer; We sincerely appreciate your excellent comment. I am sorry that this is careless mistake and this is figure 5 but not figure 4. Therefore, this is corrected.

  1. “Figure 5. Effects of FABP ligand 6 on cellular metabolic functions. Planar 2D cultured HNCPE 163 cells were not treated (Ctrl) or were treated with 10 mM FABP ligand 6 (MF6), a specific inhibitor of 164 FABP5 and FABP7, and each sample (n=6) was loaded to real-time metabolic function analysis using 165 a Seahorse XFe96 Bioanalyzer. A) Measurements of oxygen consumption rate (OCR). B) Measure- 166 ments of extracellular acidification rate (ECAR). C) Key indices of metabolic parameters. *P<0.05, 167 **P<0.01, and ***P<0.001. Oligo: oligomycin. FCCP: carbonyl cyanide p-trifluoromethoxyphenylhy- 168 drazone. R/A: a mixture of rotenone/antimycin A 2DG: 2-deoxyglucose ”Highlight non-significant statistics as 'ns' on the graph figures.

Answer; We sincerely appreciate your excellent comment. In accordance with the reviewer's comments, “ns” has been represented for indicators that are not significantly different in the revised Figure 5.

  1. “Figure 6. qPCR analysis for UPR-related factors and AQP1. 2D cultured HNPCE were subjected 206 to qPCR analysis, and the mRNA expression of UPR-related factors, including GRP78, GRP94, 207 PEAK, ATF4, IRE and CHOP, and AQP1 was estimated. Experiments were repeated three times 208 using freshly prepared cells (n=3 each) in each experiment. *P<0.05, **P<0.01. “Correct the figure graphs as follows: For AQP1: Label each bar, error bar, and statistics.§  For GRP78, GRP94, and IRE1: Label each individual bar, error bar, and statistics. §  Highlight non-significant results as 'NS'

Answer; We sincerely appreciate your excellent comment. As suggested, this figure is properly corrected.

  1. “To study the effects of FABP ligand 6 on unfolded protein responce (UPR) of HNPCE 191 cells, mRNA expression of major UPR-related factors including glucose-regulated protein 192 78 (GRP78), glucose-regulated protein 94 (GRP94), PKR-like ER kinase (PEAK), activating 193 transcription factor 4 (ATF4), inositol requiring enzyme-1 (IRE1) and C/EBP homologous 194 protein (CHOP) was evaluated by qPCR. In addition, since HNPCE cells are known as the 195 main cell responsible for the blood-aqueous barrier (BAB) of the eye [28] and aquaporin 1 196 (AQP1) and Na+/K+ ATPase control the rate of aqueous humor formation and are located 197 in the ciliary body epitheliums and the iris [29,30], the effect of FABP ligand 6 on mRNA 198 expression of AQP1 in HNPCE cells was also studied. As shown in Fig. 6, gene expression 199 of all of these UPR-related factors and AQP1 were significantly upregulated by FABP lig- 200 and 6. “Please present the results with appropriate subheadings. The results section should focus on data representation, while literature references can be included in the discussion section. Highlight the results clearly, including exact fold changes in expression compared to the control and corresponding p-values. For example, in the AQP1 qPCR bar graph, significance is not highlighted, and the axis labels are not properly formatted. Additionally, the AQP1 expression between the control and MF6 group does not appear to show a difference (MF6/Control ~1.04)

Answer; We sincerely appreciate your excellent comment. We sincerely appreciate your excellent comment. As pointed out with other comments (#7, #12, #13), we carefully revised this figure with proper p-values and axis labeling.

  1. Comments on the Quality of English Language. The manuscript would benefit from further language refinement to enhance clarity and readability.

Answer; We sincerely appreciate your excellent comment. Quality of English is carefully checked by a native English-speaking scientist and his certificate is attached.

Round 2

Reviewer 2 Report

Comments and Suggestions for Authors

The authors' response to the comments is fairly satisfactory.

In the revised manuscript, please address the following minor comments:

As shown in Fig. 6, the expression of the 209 gene and all UPR-related factors, as well as AQP1, was significantly upregulated by FABP ligand 6.

·      This statement is not correct. ATF6 and GRP94 do not appear to be significantly upregulated (no p-value or asterisk is present on the bar graph).

Figure 6. qPCR analysis for UPR-related factors and AQP1. 2D cultured HNPCE cells were subjected to qPCR analysis, and the mRNA expression of UPR-related factors, including GRP78, GRP94, PEAK, ATF4, IRE, CHOP, and AQP1, was estimated. Experiments were repeated three times using freshly prepared cells (n=3 each) in each experiment. ns: non-significant statistics, *P<0.05, and **P<0.01 (Student’s t-test).”

·      'ns: non-significant statistics' is not mentioned in any of the bar graphs in Figure 6.

Author Response

Dear Editor,

Thank you very much for the constructive comments concerning our manuscript “FABP5 is a crucial factor for the maintenance of functions of human non-pigmented ciliary epithelium cells”. We carefully checked all of the reviewer’s comments and prepared a revised version of our paper that takes these comments into account. The changes are listed below.

Reviewer 2 comment

The authors' response to the comments is fairly satisfactory.

In the revised manuscript, please address the following minor comments:

“As shown in Fig. 6, the expression of the 209 gene and all UPR-related factors, as well as AQP1, was significantly upregulated by FABP ligand 6.”

  • This statement is not correct. ATF6 and GRP94 do not appear to be significantly upregulated (no p-value or asterisk is present on the bar graph).

Answer; We sincerely appreciate your excellent comment. As according your suggestion, corresponding sentence is corrected to ‘As shown in Fig. 6, the gene expression of all these UPR-related factors except GRP94 and ATF6 as well as AQP1 was significantly upregulated by FABP ligand 6.’.

“Figure 6. qPCR analysis for UPR-related factors and AQP1. 2D cultured HNPCE cells were subjected to qPCR analysis, and the mRNA expression of UPR-related factors, including GRP78, GRP94, PEAK, ATF4, IRE, CHOP, and AQP1, was estimated. Experiments were repeated three times using freshly prepared cells (n=3 each) in each experiment. ns: non-significant statistics, *P<0.05, and **P<0.01 (Student’s t-test).”

'ns: non-significant statistics' is not mentioned in any of the bar graphs in Figure 6.

Answer; We sincerely appreciate your excellent comment. As according your suggestion, ‘ns’ is included in GRP94 and ATF6
